# Unveiling the Role of Protein Kinase C θ in Porcine Epidemic Diarrhea Virus Replication: Insights from Genome-Wide CRISPR/Cas9 Library Screening

**DOI:** 10.3390/ijms25063096

**Published:** 2024-03-07

**Authors:** Jinglin Zhou, Zhihua Feng, Deyang Lv, Duokai Wang, Kai Sang, Zhihao Liu, Dong Guo, Yangkun Shen, Qi Chen

**Affiliations:** Fujian Key Laboratory of Innate Immune Biology, Biomedical Research Center of South China, College of Life Science, Qishan Campus, Fujian Normal University, Fuzhou 350117, China; qsx20211380@student.fjnu.edu.cn (J.Z.); fengzhihua@fjnu.edu.cn (Z.F.); qsx20211379@student.fjnu.edu.cn (D.L.); qsx20211383@student.fjnu.edu.cn (D.W.); sangk2023@163.com (K.S.); shenyunyvqiong@163.com (Z.L.); guodong1102@icloud.com (D.G.)

**Keywords:** genome-scale CRISPR screen, PEDV (porcine epidemic diarrhea virus), susceptibility, PKCθ (protein kinase C θ), BOK (B-cell lymphoma 2 (BCL-2) ovarian killer), mitochondrial apoptosis, PEDV endocytosis, PEDV replication

## Abstract

Porcine epidemic diarrhea virus (PEDV), a member of the Alpha-coronavirus genus in the Coronaviridae family, induces acute diarrhea, vomiting, and dehydration in neonatal piglets. This study aimed to investigate the genetic dependencies of PEDV and identify potential therapeutic targets by using a single-guide RNA (sgRNA) lentiviral library to screen host factors required for PEDV infection. Protein kinase C θ (PKCθ), a calcium-independent member of the PKC family localized in the cell membrane, was found to be a crucial host factor in PEDV infection. The investigation of PEDV infection was limited in Vero and porcine epithelial cell-jejunum 2 (IPEC-J2) due to defective interferon production in Vero and the poor replication of PEDV in IPEC-J2. Therefore, identifying suitable cells for PEDV investigation is crucial. The findings of this study reveal that human embryonic kidney (HEK) 293T and L929 cells, but not Vero and IPEC-J2 cells, were suitable for investigating PEDV infection. PKCθ played a significant role in endocytosis and the replication of PEDV, and PEDV regulated the expression and phosphorylation of PKCθ. Apoptosis was found to be involved in PEDV replication, as the virus activated the PKCθ-B-cell lymphoma 2 (BCL-2) ovarian killer (BOK) axis in HEK293T and L929 cells to increase viral endocytosis and replication via mitochondrial apoptosis. This study demonstrated the suitability of HEK293T and L929 cells for investigating PEDV infection and identified PKCθ as a host factor essential for PEDV infection. These findings provide valuable insights for the development of strategies and drug targets for PEDV infection.

## 1. Introduction

Porcine epidemic diarrhea (PED) is a highly contagious intestinal disease caused by the PED virus (PEDV). It is characterized by symptoms such as watery diarrhea, vomiting, dehydration, and a high mortality rate in neonatal piglets [1,2]. PED was first identified in the United Kingdom in 1971, and it subsequently spread globally, affecting regions such as Europe, the Americas, and Asia [1,3,4,5]. In 2010 and 2013, there were significant outbreaks of PED that rapidly spread to swine farms in China and the United States. These outbreaks have caused significant economic losses to the swine-producing industry [5,6]. PED remains a persistent issue in the swine-producing industry globally, without any therapeutic solution [7]. PEDV is an *Alphacoronavirus* belonging to the family *Coronaviridae* of the order *Nidovirales*. It has a genome size of approximately 28 kb and is an enveloped positive-sense single-stranded RNA virus. Its genome contains at least seven open reading frames (ORFs) that encode four structural proteins (SPs): the membrane (M), envelope (E), nucleocapsid (N), and spike (S) proteins. Additionally, it encodes 16 nonstructural proteins (NSPs), 1 accessory protein (ORF3), and a 3′-untranslated region (UTR) and 5′-UTR [8].

Research on PEDV is typically conducted in vitro using Vero and porcine epithelial cell-jejunum 2 (IPEC-J2) cells. However, Vero cells are inadequate for studying host response in enteric cells because of the defects in interferon production [9]. Similarly, PEDV replication is limited in IPEC-J2 cells because of several cytokine deficiencies [10,11,12,13]. These limitations hinder further understanding of the related mechanisms underlying the viral infection. Therefore, we selected human embryonic kidney (HEK) 293T and L929 cells as appropriate cell lines for PEDV investigation because of their suitability as model cells for studying viral infection.

The clustered regularly interspaced short palindromic repeats (CRISPR)/Cas9 technique is an efficient method for inducing targeted loss-of-function mutations in diverse species [14]. In recent years, the genome-wide CRISPR knockout (GeCKO) screening strategy has emerged as a potent technology for identifying therapeutic targets in various diseases, including host factors involved in viral infection [15]. Several studies have used the GeCKO to identify the host factors of various viruses. For example, ACE2 has been identified as a host factor for SARS-CoV-2, while SLC35A1 has been identified as a host factor for porcine deltacoronavirus (PDCoV) and influenza A virus (IAVs) [16,17,18].

Protein kinase C θ (PKCθ) is a calcium-independent protein kinase found in cell membranes. It belongs to the family of serine/threonine kinases and plays a crucial role in diverse cellular processes across different cell types [19]. Specifically, the PKCθ isoform is essential for the immune system, particularly for T-cell activation, cell survival, and differentiation [20]. Recent studies have reported a role for PKCθ in viral infections. For instance, human immunodeficiency virus type 1 (HIV-1) infection increases PKCθ phosphorylation in CD4+ T cells, thereby promoting its replication [21]. Similarly, PKCθ facilitates the lytic cycle of Epstein–Barr virus (EBV) in B cells through autophagy [22]. Additionally, other isoforms of the PKC family are significant contributors to viral infection. PKCδ is used by the influenza virus for assembly and replication [23]. PKCδ high-mobility group box 1 (HMGB1) activation is necessary for porcine reproductive and respiratory syndrome virus (PRRSV) to trigger an inflammatory response [24]. Furthermore, herpes simplex virus-1 (HSV-1) can phosphorylate PKC zeta/lambda, leading to increased expression of IL-15, without affecting viral infectivity and replication [25]. However, further investigation is required to better understand the role of PKCθ in viral infection.

B-cell lymphoma 2 (BCL-2) ovarian killer (BOK) is a member of the BCL-2 family and is known for its role in triggering mitochondrial apoptosis via caspase-3 activation [26]. Studies have revealed a correlation between BOK and viral infections. For example, the SARS-CoV-2 membrane (M) protein directly binds to BOK, leading to mitochondrial apoptosis and exacerbation of lung injury [27]. Apoptosis is an evolutionarily conserved process that involves signal transduction and is associated with viral infection-induced cell death. It has been studied more extensively than necroptosis and pyroptosis [28,29]. PEDV infection induces apoptosis in target cells [30,31,32]. Although apoptosis is an innate cellular response to invading pathogens, many viruses have developed strategies to manipulate host cellular apoptosis and enhance viral replication [33,34,35]. However, the extent to which PEDV manipulates apoptosis to increase viral replication remains unclear.

In this study, we found that HEK293T and L929 cells were more susceptible to PEDV than Vero and IPEC-J2 cells. Subsequently, we conducted GECKO genetic screening in HEK293T cells to identify host factors required for PEDV infection. PKCθ was found to be a critical host factor for PEDV infection. Furthermore, PKCθ KO induced the downregulation of BOK expression and mitochondrial apoptosis. These findings provide new insights into the development of antiviral drugs or vaccines for PEDV infection.

## 2. Results

### 2.1. HEK293T and L929 Cells Are More Susceptible to PEDV Than Vero and IPEC-J2 Cells

HEK293T and L929 cells derived from human kidney and mouse fibroblasts have been extensively used in various studies. A previous study demonstrated the susceptibility of HEK293 cells to PEDV infection [36]. To further investigate the susceptibility of HEK293T and L929 cells to PEDV infection, both the cell lines were infected with PEDV at a multiplicity of infection (MOI) of 1. The susceptibility of the cells was monitored at 6, 12, 24, 36, and 48 h post-incubation (hpi) using reverse transcription real-time PCR (qRT-PCR) and Western blotting (WB). The results (Figure 1A–F) indicate that PEDV could infect both HEK293T and L929 cells, and the fold of PEDV infection increased over time. To further validate the difference in susceptibility among HEK293T, L929, Vero, and IPEC-J2 cells, the four cell types were infected with PEDV at an MOI of 1. The susceptibility of the four cell types was observed at 48 hpi using qRT-PCR and WB. The results (Figure 1G–L) indicate that PEDV susceptibility was higher in HEK293T and L929 cells than that in Vero and IPEC-J2 cells.

### 2.2. Genome-Scale CRISPR Screens Have Identified the Host Factors Associated with PEDV Infection

To identify the essential host factors for PEDV infection, we generated stable cell lines expressing Cas9 (HEK293T-Cas9) (Appendix A). A human sgRNA lentiviral library was constructed, consisting of 123,411 sgRNAs. These sgRNAs were designed and synthesized to target 20,914 genes in the human genome. The sgRNA-expressing lentivirus was transfected into HEK293T, followed by inoculation with PEDV at an MOI of 0.3. After five rounds of PEDV infection (MOI = 1), the PEDV-resistant cells were enriched (Figure 2A). The genomic DNA (gDNA) of the surviving cells was extracted and subjected to next-generation DNA sequencing (NGS) to determine the enrichment of the integrated sgRNAs (Appendix A). The genes enriched in HEK293T cells were selected for further verification using the GeCKO. Enrichment scores for PKCθ were relatively high (Figure 2B).

### 2.3. PEDV Infection Is Positively Correlated with PKCθ Expression and Phosphorylation

The innate immune system of host cells has evolved antiviral proteins that target various stages of the viral life cycle [37]. PKCθ, a host cell protein, has been implicated in viral replication and immune responses [22,38]. The expression and phosphorylation of PKCθ during PEDV infection were investigated to explore its potential antiviral function. To validate PKCθ as being necessary for PEDV infection, we used HEK293T and L929 cells for further examination. HEK293T and L929 cells were infected with PEDV at an MOI of 1 at 6, 12, 24, and 48 hpi. PKCθ expression was evaluated using qRT-PCR and Western blotting. The mRNA level of PKCθ was upregulated in PEDV-infected HEK293T and L929 cells compared with that in uninfected cells (Figure 3A,B). PKCθ expression increased in PEDV-infected HEK293T and L929 cells compared to that in uninfected cells (Figure 3C–F). Additionally, PEDV infection at an MOI of 1 induced PKCθ expression and phosphorylation in HEK293T and L929 cells at 24 and 48 hpi, as observed using Western blotting (Figure 3G–J). These findings suggest that PEDV infection induced PKCθ expression and phosphorylation in the host cells.

### 2.4. PKCθ Determines PEDV Replication

The effect of PKCθ on PEDV replication was examined by constructing PKCθ-knockout, overexpressed, and rescued overexpressed HEK293T and L929 cells. The cells were then analyzed using Western blotting (Figure 4A–D). Next, PKCθ-knockout, overexpressed, and rescued overexpressed HEK293T and L929 cells were infected with PEDV (MOI = 1) at 24, 48, and 72 hpi. RNA copies of PEDV and its associated N proteins were verified using qRT-PCR and Western blotting. Viral RNA copies and viral N protein levels were reduced in PKCθ-knockout HEK293T and L929 cells compared with uninfected cells. In contrast, the levels of viral RNA copies and viral N protein were higher in PKCθ overexpression and overexpression-rescued HEK293T and L929 cells than in uninfected cells (Figure 4E–J). These findings indicate that PKCθ regulated PEDV replication.

### 2.5. PKCθ Is Involved in PEDV Endocytosis Rather Than Absorption

The life cycle of PEDV includes absorption, endocytosis, replication, and propagation [39]. To investigate the role of PKCθ in the absorption and endocytosis of PEDV, PKCθ-knockout HEK293T and L929 cells were infected with PEDV (MOI = 1). The cells were either kept at 4 °C for 1 hpi to assess absorption or shifted to 37 °C for 1 hpi to evaluate viral entry. There were no observable differences in the RNA copies and titer of absorbed virions in PKCθ-knockout HEK293T and L929 cells compared to those in uninfected cells, as determined through qRT-PCR and TCID_50_ analyses (Figure 5A–D). However, the RNA copies of internalized virions were significantly decreased in PKCθ-knockout 293T and L929 cells compared to those in uninfected cells (Figure 5E,F). Confocal imaging also showed consistent results (Figure 5G,H). These results indicate that PKCθ is involved in PEDV endocytosis rather than absorption.

### 2.6. BOK Plays a Positive Role in PEDV Replication

RNA-seq can comprehensively analyze the genetic information of all transcripts in a biological niche. We used this powerful technology to assess transcriptome alterations in PKCθ-knockout HEK93T and L929 cells infected with PEDV (MOI = 1) for 48 hpi, in comparison to PEDV-infected HEK93T and L929 cells (MOI = 1). This study aimed to gain insights into the role of PKCθ in determining PEDV replication. To avoid sample bias, three replicates were collected for each group and used to construct RNA-seq libraries. The results show that 20,914 human genes and 20,611 mouse genes were expressed in both PEDV-infected PKCθ-knockout HEK293T and L929 cells (*n* = 6) and PEDV-infected HEK293T and L929 cells (*n* = 6). In PEDV-infected PKCθ-knockout HEK293T cells, 1699 genes were significantly upregulated, and 1433 were significantly downregulated compared to those in PEDV-infected HEK293T cells. These findings were based on the criteria of |log2 fold-change| ≥ 1 and FDR (false discovery rate) ≤ 0.05 (Figure 6A,C). In addition, 132 significantly upregulated and 217 downregulated genes were identified in PEDV-infected PKCθ-knockout L929 cells compared to PEDV-infected L929 cells (|log2 fold change| ≥ 1 and FDR ≤ 0.05; Figure 6B,D). Subsequently, the overlapping DEGs including a total of 6 upregulated and a total of 27 downregulated genes from PKCθ-knockout PEDV-infected HEK93T and L929 cells compared to PEDV-infected HEK93T and L929 cells were analyzed and are summarized in Table 1 and Table 2. The top four genes with the highest downregulated and upregulated mRNA levels were BOK (downregulation), translocator protein of 18 kDa (TSPO; downregulation), MT-ND2 (upregulation), and MT-ATP8 (upregulation), respectively. Moreover, to better understand the functions of the DEGs, KEGG pathway enrichment analyses were performed in PKCθ-knockout PEDV-infected HEK93T and L929 cells compared to those in PEDV-infected HEK93T and L929 cells. KEGG pathway classification cluster analysis divided the DEGs into five categories: cellular processes, environmental information processing, genetic information processing, metabolism, and organismal systems (Figure 6E,F). These five categories included 20 subcategories. Most genes were related to cell growth and death in the cellular processes category and to signal transduction in the environmental information processing category. KEGG pathway analysis was performed to evaluate the biological and ontological significance of the DEGs (Figure 6G,H). The results show that DEGs were involved in a variety of biological activities, and more genes were annotated as being related to RNA degradation, pentose phosphate pathway, methane metabolism, glycolysis, fructose, mannose metabolism, carbon metabolism, biosynthesis of amino acids, and apoptosis. Notably, BOK is a crucial modulator of mitochondrial apoptosis, which enters into mitochondria causing mitochondrial stress, and then induces the leakage of cytochrome c from mitochondria to activate caspase-9 and bind with the apoptotic protease activating factor-1 (Apaf-1), leading to the formation of apoptosome. Subsequently, apoptosomes recognize and activate caspase-3, and then activated caspase-3 is transported into the nucleus, resulting in the cleavage of PARP1 and the loss of cell homeostasis, leading to apoptosis [26,28,40]. These results suggest that PKCθ might determine PEDV replication via BOK and mitochondrial apoptosis.

### 2.7. BOK Affects the Mitochondrial Apoptotic Pathway in PEDV-Infected Cells

To validate the quality of the RNA-seq data, BOK expression was validated in PKCθ-knockout PEDV-infected HEK293T and L929 cells (MOI = 1) at 48 hpi compared to that in PEDV-infected HEK293T and L929 cells (MOI = 1) using qRT-PCR and Western blot analysis (Figure 7A,B). The results indicate that the levels of RNA copies and BOK proteins were reduced in PEDV-infected PKCθ-knockout HEK293T and L929 cells compared to the control group. The levels of leaked cytochrome c, BOK, cleaved-caspase9, pro-caspase 3, cleaved-caspase 3, and cleaved PARP1 were examined in PKCθ-knockout PEDV-infected HEK293T and L929 cells (MOI = 1) at 48 hpi compared to the control group using confocal imaging and Western blotting (Figure 7C–G). The results indicate that PEDV-infected PKCθ-knockout HEK293T and L929 cells had decreased levels of these proteins compared to the control group. In addition, to assess changes in apoptotic signal intensity, Annexin V-FITC was used to analyze the percentage of apoptotic cells in PKCθ-knockout PEDV-infected HEK293T and L929 cells (MOI = 1) at 48 hpi compared to PEDV-infected HEK293T and L929 cells (MOI = 1). The percentage of apoptotic cells was determined using Annexin V-FITC analysis via flow cytometry (FACS) (Figure 7H). The proportion of apoptotic cells in PEDV-infected PKCθ-knockout HEK293T and L929 cells was lower than that in the control group. The irreversible loss of mitochondrial membrane potential (ΔΨm) is a significant event in early apoptotic cells [41]. FACS analysis of JC-1 staining revealed a higher proportion of cells with low ΔΨm in the control group than in the PEDV-infected (MOI = 1) PKCθ-knockout HEK293T and L929 cells at 48 hpi (Figure 7I). Additionally, in this study, cleaved caspase 3 was detected in PKCθ-knockout PEDV-infected HEK293T and L929 cells (MOI = 1) at 48 h. Cleaved-caspase 3 activity was lower in PKCθ-knockout PEDV-infected HEK293T and L929 cells than in PEDV-infected HEK293T and L929 cells at 48hpi (Figure 7J). These results indicate that PKCθ knockout affected BOK expression and mitochondrial apoptosis.

### 2.8. The PKCθ-BOK Axis Is Essential for PEDV Replication via Mitochondrial Apoptosis

To confirm the effects of PKCθ on PEDV replication via BOK, we first constructed BOK-rescued expression in PKCθ-knockout HEK293T and L929 cells (Figure 8A–C). Next, we assessed the effect of BOK-rescued overexpression in PKCθ-knockout PEDV-infected HEK293T and L929 cells (MOI = 1) at 48 h and compared it with that in PEDV-infected HEK293T and L929 cells, as well as PKCθ-knockout PEDV-infected HEK293T and L929 cells. This was performed using qRT-PCR and Western blot analysis (Figure 8D–H). The results reveal that the levels of RNA copies and virions were significantly higher than those in the control groups. The levels of pro-caspase 3 and cleaved-caspase 3 were assessed using Western blot analysis (Figure 8F–H). The results reveal that the level of pro-caspase 3 was reduced, while the level of cleaved-caspase 3 was increased compared to that in the control groups. Furthermore, the ratio of apoptotic cells to cells with low ΔΨm was assessed using Annexin V-FITC and JC-1 and analyzed using flow cytometry (Figure 8I,J). The results indicate higher percentages of apoptotic cells and cells with low ΔΨm compared to those in the control groups. These data suggest that the PKCθ–BOK axis played a crucial role in PEDV replication via mitochondrial apoptosis.

## 3. Discussion

PEDV is a highly virulent, re-emerging enteric coronavirus that causes severe diarrhea, vomiting, and high mortality rates in neonatal piglets [2]. Over the past few decades, PEDV has caused significant economic losses in many countries. Therefore, investigating the mechanisms of viral infection is crucial in the swine industry. Previous research on PEDV lacks in-depth and systematic investigation, particularly in relation to the aspects of receptors and co-receptors, as well as investigations into infection and replication mechanisms.

Vero and IPEC-J2 cell lines are frequently used in PEDV research. However, these cell lines exhibit limitations in terms of their immune response and viral replication. Several studies have identified additional cell lines that are susceptible to PEDV infection to address these limitations and to enhance our understanding of PEDV infection. These include rat crypt epithelial cells, primary bovine mesenchymal cells, and IP2-2I cells [10,42,43]. Studies have shown that HEK293 cells are susceptible to PEDV infection [36]. In the present study, we assessed the susceptibility of HEK293T and L929 cells to PEDV infection, based on their robust growth, stable cytokine expression, and efficient transfection capabilities. The results show that both HEK293T and L929 cells were susceptible to PEDV infection, suggesting that these cell models are ideal for investigating PEDV infection in vitro. Additionally, these findings imply that PEDV could pose a potential threat to mammals, especially humans. Notably, receptors and co-receptors required for PEDV infection may have discrepancies among different host cells, raising questions about the validation of these receptors and co-receptors in in vivo studies. We illustrated PEDV endocytosis and replication relies on PKCθ-BOK-caspase3 mitochondrial apoptosis in Figure 9. 

Genome-wide CRISPR-Cas9 gene disruption screening is a powerful technique for identifying host factors required for viral infection. Our screening results identify PKCθ as a host factor involved in the reactivation and transcription of HIV-1 and the facilitation of the lytic cycle of EBV [18,22]. In addition, PKCθ plays multiple roles in the activation of signaling pathways, including the activation of nuclear factor κB (NF-κB) and nuclear factor of activated T-cells (NFAT) and activating protein 1 (AP-1) [44,45]. Additionally, OR52M1, GPI, MGEA5, ELTD1, SPAG8, GNG7, ELTD1, C14ORF159, METTL17, and TMCO5A have been identified as host factors of PEDV. SPAG8 has been investigated as a potential host factor for human papillomavirus (HPV) infection [46]. GNG7 and METTL17 have been identified as potential host factors for the infection of cyprinid herpesvirus 2 (CyHV-2) and HIV-1 [47,48]. We further analyzed the role of these host factors and their underlying mechanisms in PEDV-infected individuals to identify potential therapeutic targets.

Apoptosis is a conserved intracellular process that acts as a defense mechanism by eliminating virus-infected cells, thereby inhibiting viral replication. Early studies demonstrated that PEDV ORF3 can promote viral proliferation by inhibiting apoptosis in infected cells [49]. In addition, PEDV Nsp6 interacts with glucosyltransferase Rab-like GTPase activator and myotubularin domain containing 4 (GRAMD4), a pro-apoptotic protein, to inhibit the apoptosis of infected cells and enhance PEDV replication [50]. However, most viruses have evolved strategies to manipulate apoptosis to replicate and release virions. For example, IAVs have developed corresponding viral structural or non-structural proteins that can manipulate pro-apoptotic signaling pathways to enhance viral replication and transmission [50]. The structural NS2B-3 protein of Japanese encephalitis virus (JEV) can induce apoptosis to facilitate the release and spread of viral particles by causing AXL degradation [51]. Furthermore, research has shown that PKCθ is crucial in triggering apoptosis and cell death [52,53]. Previous studies focusing on the mechanism of PEDV infection in cells have not determined whether PEDV replication relies on apoptosis or not. Our study provides novel insights into the ability of PEDV to induce apoptosis during replication via PKCθ. Balancing cell survival and death during PEDV infection remains a pressing issue that requires immediate resolution.

## 4. Materials and Methods

### 4.1. Cell Lines, Culture Conditions, and Viruses

HEK293T (CRL-3216), L929 (CCL-1), Vero (CCL-81), and IPEC-J2 (CCL-92) cell lines were derived from the American Type Culture Collection (ATCC) and purchased from the Cell Bank of the Chinese Academy of Sciences (Shanghai, China), and their authenticity was verified using short tandem repeat analysis. These cells were cultured in a medium (Hyclone, Seattle, WA, USA) supplemented with 10% fetal bovine serum (FBS; PAN-Biotech, Germany) and 1% penicillin-streptomycin (Hyclone) at 37 °C in a 5% CO_2_ incubator with a passaging time of <6 months. The PEDV CV777 strain (GenBank accession no: AF353511) was used in this study. The PEDV strain CV777 was subsequently cultured and quantified in HEK293T and L929 cells in DMEM supplemented with 10 μg/mL trypsin and 10% FBS (PAN-Biotech, Aidenbach, Germany) at 37 °C with 5% CO_2_.

### 4.2. HEK293T Library Generation and PEDV Screening

HEK293T-GeCKO libraries were created using the lentiGuide-Puro vector system for Cas9 and sgRNA delivery. First, HEK293T cell clones stably expressing Cas9 components (HEK293T-Cas9) were generated using lentivirus transfection of the Cas9 transgene. Next, HEK293T cells were transfected with a lentivirus human gRNA library targeting 20,914 genes, using an MOI of 0.3 to ensure the delivery of a single viral construct to most cells. After 48 hpi, transfected HEK293T-Cas9 cells were selected by culturing with 2 μg/mL and 10 μg/mL puromycin (Hyclone, Seattle, WA, USA), respectively, for 2 to 3 days. The cells were then infected with PEDV at an MOI of 0.3. After 24–30 hpi, the cells were washed with PBS (Hyclone, Seattle, WA, USA) to remove dead cells. Subsequently, fresh medium was added to the surviving clones. The cells were cultured in DMEM (Hyclone, Seattle, WA, USA) supplemented with 10% FBS (PAN-Biotech, Aidenbach, Germany) and 1% penicillin-streptomycin solution (BasalMedia, Shanghai, China), and they were subjected to five successive rounds of PEDV infection. The cells became resistant to viral infection after these rounds of infection. At the end of the screening, the surviving resistant cells were expanded and analyzed using deep sequencing analysis by Sangon Biotech Company (Shanghai, China).

### 4.3. Generation of KO, Overexpression, and Rescued Overexpression Cell Lines

We constructed a plasmid, PX459, containing sgRNA-1 and sgRNA-2 targeting the PKCθ gene, which was then transfected into HEK293T cells. Additionally, we constructed a LentiV2 plasmid containing sgRNA-3 and sgRNA-4 targeting the PKCθ gene, which was transfected into L929 cells. The sgRNA molecules that were used are listed in Table 3. Transfection was performed on HEK293T and L929 cells, followed by selection using puromycin (Beyotime, Shanghai, China) at concentrations of 1 and 10 μg/mL, respectively, for 3 days. The surviving cells were cultured in a medium containing 10% FBS (PAN-Biotech, Aidenbach, Germany). To generate homogeneous PKCθ-KO cell lines, HEK293T and L929 cells were subcloned using a finite continuous dilution in 96-well plates (NEST Biotech, Shanghai, China) for clonal expansion. The successful construction of PKCθ-KO HEK293T and L929 cells was validated using Western blotting (WB). PKCθ-overexpressing HEK293T and L929 cells were generated. Additionally, PKCθ-rescued overexpressing HEK293T and L929 cells and BOK-rescued overexpressing PKCθ-knockout HEK293T and L929 cells were generated. Fragments of PKCθ and BOK coding sequences (CDS) were synthesized and subcloned into the PLVX-puro vector by the General Biology Company (Hefei, Anhui, China). The CDS and packaging plasmids were co-transfected into HEK293T cells using Lipofectamine 2000 (Invitrogen, Waltham, MA, USA) for 48 hpi. Subsequently, the supernatant was collected and used to infect the HEK293T and L929 cells. This process was repeated at least thrice. After infection, puromycin selection was performed at concentrations of 1 and 10 μg/mL for HEK293T and L929 cells, respectively.

### 4.4. RNA Extraction and Quantitative RT-PCR

Total RNA was extracted using TRIzol reagent (TaKaRa, Beijing, China), and its concentration, purity, and integrity were evaluated using a NanoDrop spectrophotometer (Thermo scientific, Waltham, MA, USA). Reverse transcription was performed using 1 μg of total RNA with HiScript II Q RT SuperMix for qPCR (+gDNA wiper) and gDNA eraser (Vazyme, Shanghai, China). The concentration, purity, and integrity of cDNA were evaluated using a NanoDrop spectrophotometer (Thermo scientific, Waltham, MA, USA). The qRT-PCR measurements were conducted using SYBR Green MasterMix (11203ES50, YEASEN, Shanghai, China) and StepOne Software v.2.3 (Applied Biosystems, Carlsbad, CA, USA) with 40 cycles (three biological replicates). Data were analyzed using the ΔΔCt (cycle threshold) method and normalized to the expression of the reference gene, *GAPDH*. The primer sequences used for qRT-PCR are listed in Table 4.

### 4.5. Western Blot

Following various treatments, the cells were washed with ice-cold PBS (Hyclone, Seattle, WA, USA) and then collected through gentle scraping. Total proteins were extracted by lysing the cells with radioimmunoprecipitation assay (RIPA) lysis buffer (Beyotime, Shanghai, China) supplemented with a phosphatase inhibitor cocktail (Beyotime, China) and protease inhibitor cocktail (Beyotime, China). The resulting cell lysates were centrifuged at 14,000× *g* for 15 min at 4 °C. Subsequently, they were denatured for 10 min in a 5× SDS-PAGE loading buffer (Beyotime, China). Proteins were separated through SDS-PAGE and subsequently transferred to polyvinylidene fluoride (PVDF, Beyotime, China) for Western blot analysis. The membranes were blocked with NcmBlot blocking buffer (NCM Biotech, Suzhou, China) for 10 min. Subsequently, the membranes were incubated with primary antibodies for 8 h at 4 °C. The primary antibodies used are listed in Table 5, which were diluated in 5% BSA (Solarbio, Beijing, China). The membranes were then incubated with secondary antibodies (ThermoFisher, Waltham, MA, USA) which were diluated in WB secondary antibody diluent solution (Beyotime, Shanghai, China) with 1:1000 dilution ratio for 2 h at room temperature. Proteins were detected using an enhanced chemiluminescence (ECL) substrate (Thermo Fisher, Waltham, MA, USA). Protein production was quantified by analyzing the band densities of the target proteins using the ImageJ software version 1.57. The analysis was based on the density value relative to GAPDH protein. 

### 4.6. TCID50 and Plaque Assays

Viral titers were quantified using 96-well tissue culture plates (NEST Biotech, China) and 50% tissue culture infective dose (TCID_50_) assays. Infected cell samples were collected and serially diluted 10-fold; 100 μL of each dilution was added to a single well of a 96-well plate, and 800 μL of each dilution was added to a single well of a 6-well plate. The plates were then incubated at 37 °C and 5% CO_2_ for 2–3 days, during which the viral cytopathic effect (CPE) was monitored. Viral titers were determined using the Spearman–Karber method and expressed as TCID50 per mL. Plaques were visually observed and stained with 0.1% crystal violet for further confirmation.

### 4.7. Indirect Immunofluorescence (IF) Staining

Cells were washed twice with PBS (Hyclone, UT, USA). Subsequently, cells were fixed with 4% paraformaldehyde for 30 min. The cells were stained using the Cell Plasma Membrane Staining Kit with DiO (C1993S, Beytoime, China), following the manufacturer’s instructions. The fixed cells were subsequently washed and blocked using blocking buffer [0.15% (*v*/*v*) Triton X-100 and 5% (*m*/*v*) BSA in PBS] for 30 min. The cells were immunolabeled using mouse anti-PEDV N (1:200, SD-2-1, Medgenes, Brookings, SD, USA) primary antibody and incubated overnight at 4 °C to detect PEDV endocytosis. Additionally, cells were incubated overnight at 4 °C with rabbit anti-cytochrome C (1:100, ab133504, Abcam, Cambridge, UK) and mouse anti-TOM20 (1:150, sc-17764, Santa Cruz Bio, Dallas, TX, USA) antibodies to detect the leakage of cytochrome c from the mitochondria. The cells were washed with PBS (Hyclone, UT, USA) and stained with Alexa Fluor 488- (1:1000, Thermo Fisher Scientific, Waltham, MA, USA), Alexa Fluor 594- (1:1000, Thermo Fisher Scientific, USA), or Alexa Fluor 647-secondary antibodies (1:1000, #AF647, Bioss, Beijing, China) at room temperature for 1 h. Nuclear staining was performed using DAPI (C1002; Beyotime, China). Images were captured using an LSM880 Zeiss confocal imaging system (63× magnification). The co-localization of PEDV with the cell membrane and cytochrome c with mitochondria was analyzed using Zeiss Zen Blue software version 3.3. A localization index of 100% indicated that PEDV-N and the cell membrane, detected by both channels, exhibited identical distribution patterns, indicating an optimal state in healthy cells.

### 4.8. Ribonucleic Acid Sequencing (RNA-Seq)

Total RNA was extracted using TRIzol reagent (T9108; TaKaRa, China), and its concentration, purity, and integrity were evaluated using a NanoDrop spectrophotometer (Thermo scientific, USA). The library was constructed by the Sangon Biotech Company (Shanghai, China). Briefly, the library-generation process involved the purification of mRNA from total RNA using Poly-T oligo-attached magnetic beads. The purified mRNA was then fragmented in an Illumina proprietary fragmentation buffer containing divalent cations at a high temperature. Subsequently, first-strand cDNA was synthesized using SuperScript II and random oligos. DNA polymerase I and RNase H were used for the second-strand cDNA synthesis. Exonuclease/polymerase activity was used to convert the remaining overhangs into blunt ends, followed by enzyme removal. Adenylated Illumina PE adapter oligos were ligated to 30 ends of the DNA fragments for hybridization. The libraries were purified using the Beckman Coulter AMPure XP system to obtain cDNA fragments measuring 400–500 bp in length. DNA fragments with adapter molecules at both ends were selectively amplified in a 15-cycle PCR using the Illumina PCR Primer Cocktail. Subsequently, the products were purified using an AMPure XP system. Quantification was performed using the Agilent high-sensitivity DNA assay on a Bioanalyzer 2100 system. Finally, the samples were sequenced on the Illumina NovaSeq 6000 platform and subjected to additional analyses by the Sangon Biotech Company, including expression differences, enrichment, and clustering (Shanghai, China).

### 4.9. Apoptosis Assay

The apoptosis assay was conducted using the Annexin V-FITC/PI Detection Kit (A211-01, Vazyme, China) following the manufacturer’s instructions. Briefly, a cell suspension containing 2 × 10^5^ cells was prepared in a binding buffer. The cells were then stained with 10 μL Annexin V-FITC and 10 μL PI for 10 min. The samples were analyzed using flow cytometry (Novocyte, Agilent, Santa Clara, CA, USA). The FACS data were analyzed using the software (Flowjo 10.8.1) included with the instrument.

### 4.10. Mitochondrial Membrane Potential (ΔΨm) Assay

Variations in ΔΨm were detected using JC-1 staining (C1073S, Beyotime, China). Specifically, 2 × 10^5^ cells were seeded in a 12-well plate (Nest Biotechnology, Wuxi, China). JC-1 staining was conducted 48 h post-infection with PEDV using 10 nM JC-1, following the manufacturer’s instructions. The cells were washed with PBS (Hyclone, UT, USA), and the ratio of aggregated JC-1 (red fluorescence; ECD channel) to monomeric JC-1 (green fluorescence, FITC channel) was measured using flow cytometry. The reduction in ΔΨm was determined by analyzing the red/green fluorescence intensity ratio. The FACS data were analyzed using the software (Flowjo 10.8.1) included with the instrument.

### 4.11. Caspase 3 (CASP) Activity Assay

CASP 3 activity in living cells was evaluated using the GreenNuc™ CASP-3 Activity Kit (C1073S, Beyotime, China) following the manufacturer’s instructions. Cells were cultured in 12-well plates (NEST Biotechnology, China) at 37 °C with 5% CO_2_ for 48 hpi to detect caspase 3 activity. Following incubation, the cells were treated with 200 μL PBS (Hyclone, UT, USA) containing 10 μM active CASP 3 detection reagents for 30 min at room temperature. The fluorescence intensity of CASP 3 activity was quantified through flow cytometry (Agilent, Novocyte, Santa Clara, CA, USA). The FACS data were analyzed using the software (Flowjo 10.8.1) included with the instrument.

### 4.12. Statistical Analysis

All experiments were performed at least three times independently. Statistical differences were analyzed using Student’s t-test or one-way analysis of variance (ANOVA), and the results were visualized using GraphPad Prism software (version 8.0). In the figures, significant differences are indicated with asterisks. “ns” indicates no significant difference. The significance levels are as follows: “*”, *p* ≤ 0.05; “**”, *p* ≤ 0.01; “***”, *p* ≤ 0.001; “****”, *p* ≤ 0.0001.

## 5. Conclusions

In conclusion, we have demonstrated for the first time that HEK293T and L929 cells are susceptible to PEDV infection, making them suitable cell models for studying PEDV. Next, we conducted genome-wide CRISPR knockout library screening of HEK293T cells. Through this screening, we identified PKCθ as a critical host factor for PEDV infection. In addition, we validated the significance of this host factor in L929 cells. Notably, PKCθ plays a significant role in PEDV endocytosis and replication via BOK-induced mitochondrial apoptosis.

## Figures and Tables

**Figure 1 ijms-25-03096-f001:**
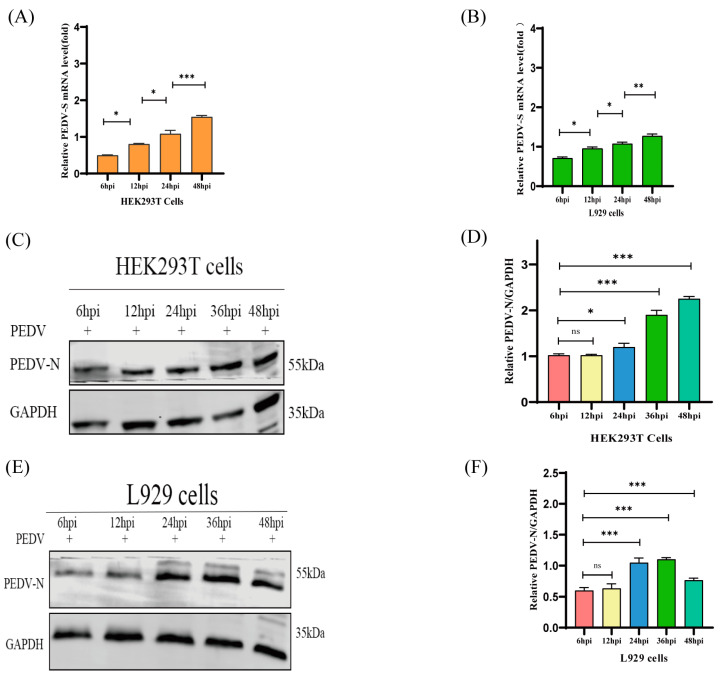
Susceptibility of different cell lines to PEDV infection. HEK293T and L929 cells demonstrate higher susceptibility to PEDV infection than Vero and IPEC-J2 cells. (**A**) The infectivity of PEDV (MOI = 1) in HEK293T cells was analyzed by measuring PEDV-S RNA levels at 6, 12, 24, 36, and 48 hpi using qRT-PCR. (**B**) The infectivity of PEDV (MOI = 1) in L929 cells was analyzed by measuring the levels of PEDV-S RNA at 6, 12, 24, 36, and 48 hpi using qRT-PCR. (**C**,**D**) Western blot analysis of the infectivity of PEDV (MOI = 1) in HEK293T cells by measuring the levels of PEDV-N protein at 6, 12, 24, 36, and 48 hpi; changes in the PEDV-N/GAPDH ratio were quantified using ImageJ software version 1.57 and plotted. (**E**,**F**) Western blot analysis of the infectivity of PEDV (MOI = 1) in L929 cells by measuring the levels of PEDV-N protein at 6, 12, 24, 36, and 48 hpi; changes in the PEDV-N/GAPDH ratio were quantified using ImageJ software version 1.57 and plotted. (**G**,**H**) qRT-PCR analysis of PEDV-S RNA levels at 48 hpi to determine susceptibility to PEDV infection (MOI = 1) in HEK293T, L929, Vero, and IPEC-J2 cells. (**I**,**J**) Western blot analysis of the levels of PEDV-N protein at 48 hpi to determine susceptibility to PEDV infection (MOI = 1) in HEK293T, Vero, and IPEC-J2 cells; changes in the PEDV-N/GAPDH ratio were quantified using ImageJ software version 1.57 and plotted. (**K**,**L**) Western blot analysis of the levels of PEDV-N protein at 48 hpi to determine the susceptibility to PEDV infection (MOI = 1) in L929, Vero, and IPEC-J2 cells; changes in the PEDV-N/GAPDH ratio were quantified using ImageJ software version 1.57 and plotted. Data are presented as mean ± standard error (SE). “*” means *p* ≤ 0.05; “**” means *p* ≤ 0.01; “***” means *p* ≤ 0.001; “ns” means not significant.

**Figure 2 ijms-25-03096-f002:**
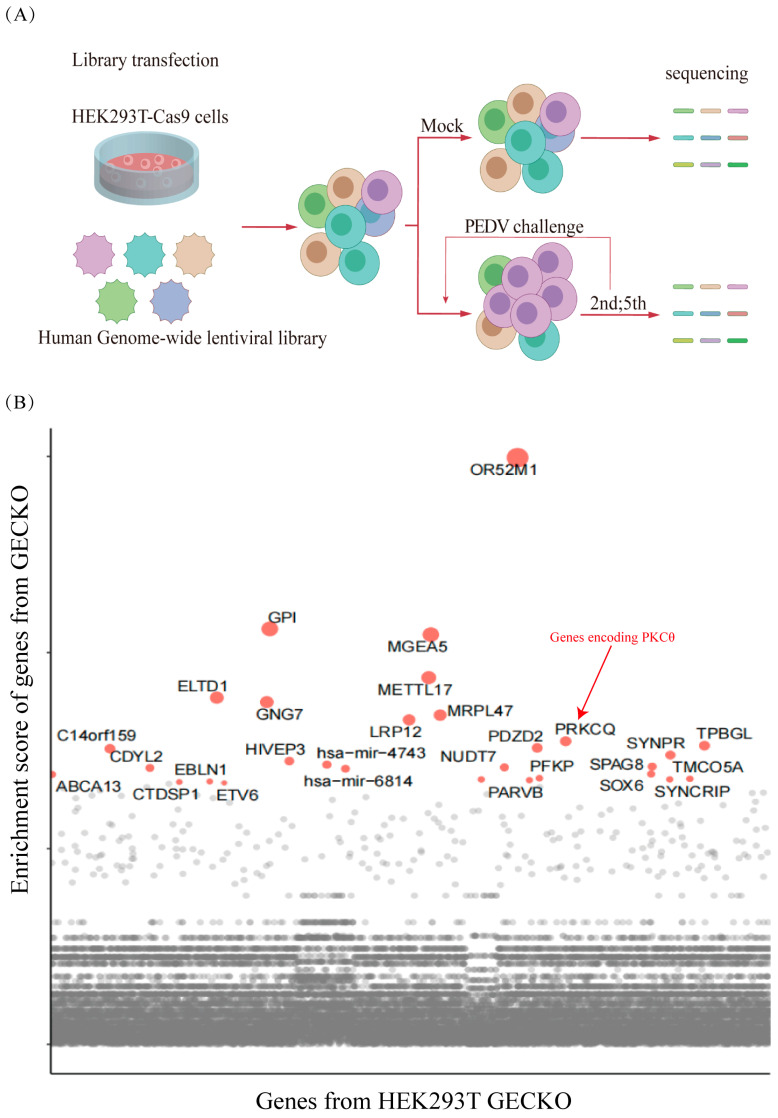
Genome-scale CRISPR screening of host factors associated with PEDV infection. (**A**) Schematic illustrating the workflow of the sgRNA lentiviral library. (**B**) Enrichment scores of the top 30 host factors in libraries.

**Figure 3 ijms-25-03096-f003:**
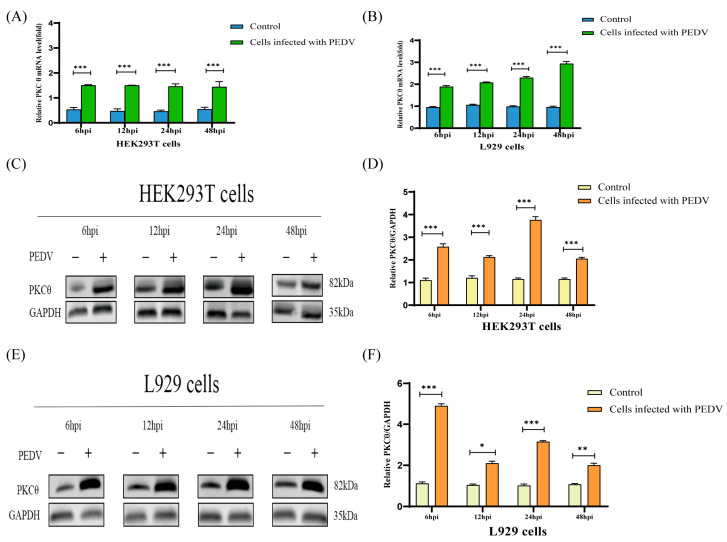
PEDV infection increases PKCθ expression and phosphorylation in HEK293T and L929 cells. (**A**) qRT-PCR analysis of PKCθ mRNA levels in HEK293T and PEDV-infected HEK293T cells (MOI = 1) at 6, 12, 24, and 48 hpi. (**B**) qRT-PCR analysis of PKCθ mRNA levels in L929 and PEDV-infected L929 cells (MOI = 1) at 6, 12, 24, and 48 hpi. (**C**,**D**) Western blot analysis of PKCθ protein levels in HEK293T and PEDV-infected HEK293T cells (MOI = 1) at 6, 12, 24, and 48 hpi; changes in the PKCθ/GAPDH ratio were quantified using ImageJ software version 1.57 and plotted. (**E**,**F**) Western blot analysis of PKCθ protein levels in L929 and PEDV-infected L929 cells (MOI = 1) at 6, 12, 24, and 48 hpi; changes in the PKCθ/GAPDH ratio were quantified using ImageJ software version 1.57 and plotted. (**G**,**H**) Western blot analysis of phosphorylated PKCθ protein levels in HEK293T and PEDV-infected HEK293T cells (MOI = 1) at 24 and 48 hpi; changes in the phosphorylated PKCθ/GAPDH ratio were quantified using ImageJ software version 1.57 and plotted. (**I**,**J**) Western blot analysis of phosphorylated PKCθ protein levels in L929 and PEDV-infected L929 cells (MOI = 1) at 24 and 48 hpi; changes in the phosphorylated PKCθ/GAPDH ratio were quantified using ImageJ software version 1.57 and plotted. “*” means *p* ≤ 0.05; “**” means *p* ≤ 0.01; “***” means *p* ≤ 0.001.

**Figure 4 ijms-25-03096-f004:**
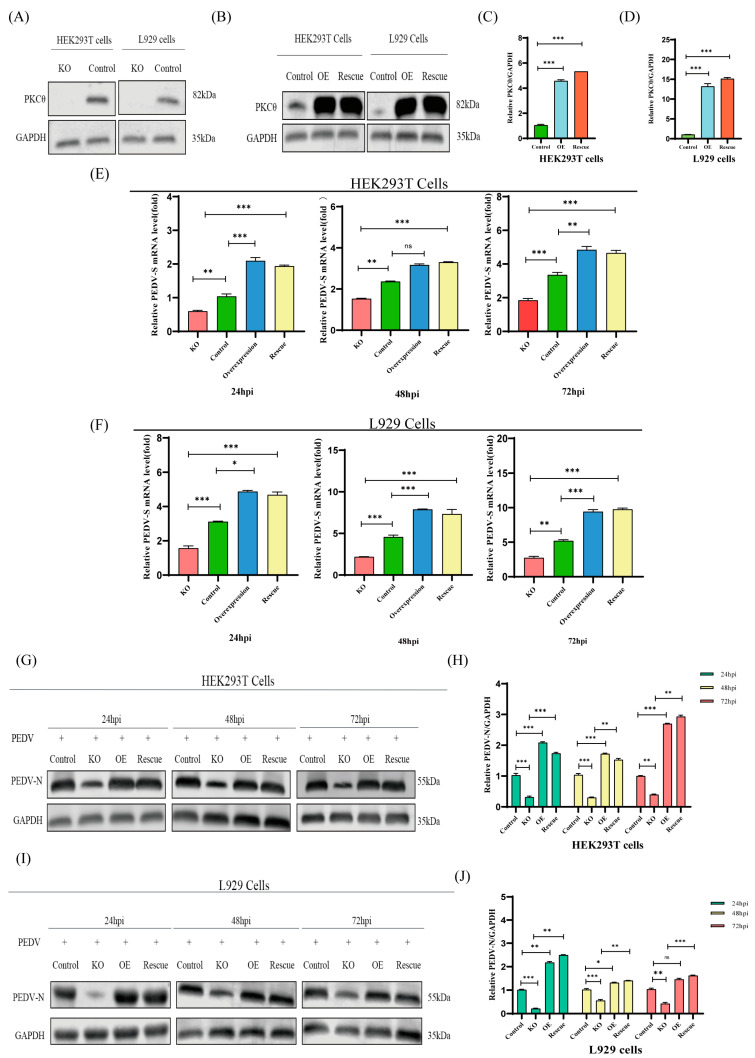
Knockout or overexpression and rescued overexpression of PKCθ inhibits or facilitates PEDV replication, respectively. (**A**) Western blot analysis of PKCθ-knockout HEK293T and L929 cells. (**B**–**D**) Western blot analysis of overexpression and rescued overexpression of PKCθ; changes in the PKCθ/GAPDH ratio were quantified using ImageJ software version 1.57 and plotted. (**E**) qRT-PCR analysis of PEDV-S RNA levels in PKCθ-knockout PEDV-infected HEK293T cells (MOI = 1), PEDV-infected HEK293T cells (MOI = 1), PKCθ-overexpressing PEDV-infected HEK293T cells (MOI = 1), and PKCθ-rescued overexpressing PEDV-infected HEK293T cells (MOI = 1) at 24, 48, and 72 hpi. (**F**) qRT-PCR analysis of the levels of PEDV-S RNA in PKCθ-knockout PEDV-infected L929 cells (MOI = 1), PEDV-infected L929 cells (MOI = 1), PKCθ-overexpressing PEDV-infected L929 cells (MOI = 1), and PKCθ-rescued overexpressing L929 cells (MOI = 1) at 24, 48, and 72 hpi. (**G**,**H**) Western blot analysis of PEDV-N protein levels in PKCθ-knockout PEDV-infected HEK293T cells (MOI = 1), PEDV-infected HEK293T cells (MOI = 1), PKCθ-overexpressing PEDV-infected HEK293T cells (MOI = 1), and PKCθ-rescued overexpressing HEK293T cells (MOI = 1) at 24, 48, and 72 hpi; changes in the PEDV-N/GAPDH ratio were quantified using ImageJ software version 1.57 and plotted. (**I**,**J**) Western blot analysis of PEDV-N protein levels in PKCθ-knockout PEDV-infected L929 cells (MOI = 1), PEDV-infected L929 cells (MOI = 1), PKCθ-overexpressing PEDV-infected L929 cells (MOI = 1), and PKCθ-rescued overexpressing PEDV-infected L929 cells (MOI = 1) at 24, 48, and 72 hpi; changes in the PEDV-N/GAPDH ratio were quantified using ImageJ software version 1.57 and plotted. “*” means *p* ≤ 0.05; “**” means *p* ≤ 0.01; “***” means *p* ≤ 0.001, “ns” means not significant.

**Figure 5 ijms-25-03096-f005:**
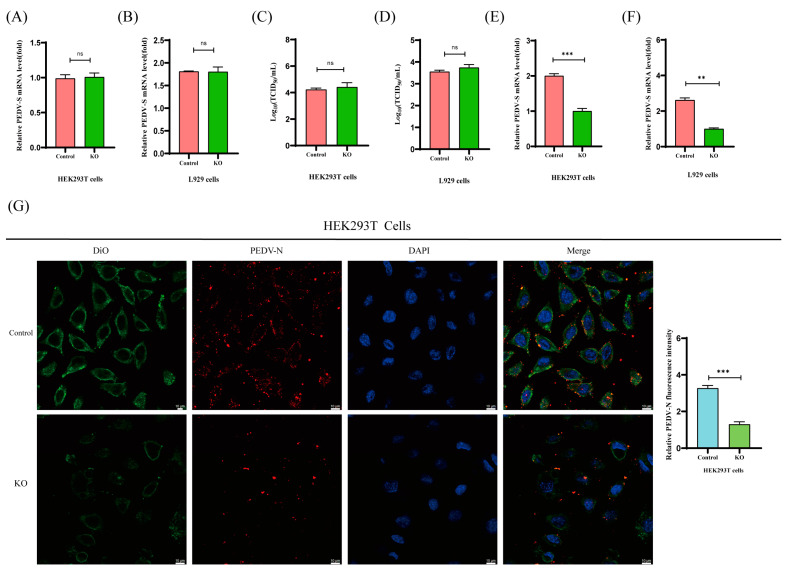
PKCθ regulates the endocytosis of PEDV rather than its absorption. (**A**,**B**) qRT-PCR analysis of PEDV-S RNA levels in PKCθ-knockout PEDV-infected HEK293T cells (MOI = 1), PEDV-infected HEK293T cells (MOI = 1), PKCθ-knockout PEDV-infected L929 cells (MOI = 1), and PEDV-infected L929 cells (MOI = 1) after inoculation for 1 h at 4 °C to allow for virus attachment without internalization. (**C**,**D**) Changes in viral titers (TCID50) of attached PEDV viral particles in PKCθ-knockout PEDV-infected HEK293T cells (MOI = 1), PEDV-infected HEK293T cells (MOI = 1), PKCθ-knockout PEDV-infected L929 cells (MOI = 1), and PEDV-infected L929 cells (MOI = 1). (**E**,**F**) qRT-PCR estimation of viral entry into PKCθ-knockout PEDV-infected HEK293T cells (MOI = 1), PEDV-infected HEK293T cells (MOI = 1), PKCθ-knockout PEDV-infected L929 cells (MOI = 1), and PEDV-infected L929 cells (MOI = 1) after inoculation for 1 h at 4 °C and then shifting to 37 °C for 1 hpi. (**G**) The entered viral particles of PEDV were also determined through colocalization of DiO-stained cell membranes (green), PEDV-N (red), and DAPI-nuclear staining (blue). Images were acquired using a Zeiss 880 inverted confocal microscope with a 63× oil-immersion objective. Scale bar, 10 μm. The colocalization of PEDV-N (red) and cell membranes (green) was analyzed. (**H**) Estimation of viral entry into PKCθ-knockout PEDV-infected L929 cells (MOI = 1) and PEDV-infected L929 cells (MOI = 1) following inoculation for 1 h at 4 °C and then shifting to 37 °C for 1 hpi. The entered viral particles of PEDV were determined through colocalization of DiO-stained cell membranes (green), PEDV-N (red), and DAPI-nuclear staining (blue). Images were acquired using a Zeiss 880 inverted confocal microscope with a 63× oil-immersion objective. Scale bar, 10 μm. The colocalization of PEDV-N (red) and cell membranes (green) was analyzed. “**” means *p* ≤ 0.01; “***” means *p* ≤ 0.001; “ns” means not significant.

**Figure 6 ijms-25-03096-f006:**
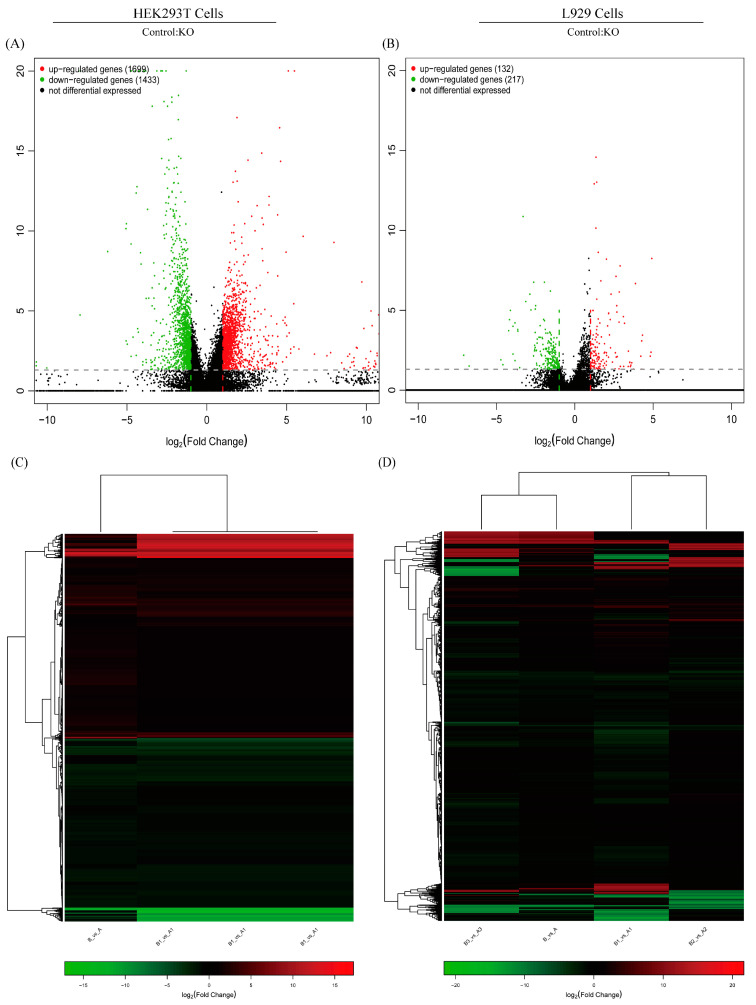
RNA-seq analysis reveals that BOK expression is decreased in PKCθ-knockout PEDV-infected HEK293T and L929 cells. (**A**) Volcano chart of DEGs in PKCθ-knockout PEDV-infected HEK293T cells (MOI = 1) and PEDV-infected HEK293T cells (MOI = 1) at 48 hpi. The *x*–*y* axis represents the log_10_-transformed gene expression levels; red, green, and black represent upregulated, downregulated, and non-DEGs, respectively. (**B**) Volcano chart of DEGs in PKCθ-knockout PEDV-infected L929 cells (MOI = 1) and PEDV-infected L929 cells (MOI = 1) at 48 hpi. The *x*–*y* axis represents the log_10_-transformed gene expression levels; red, green, and black represent upregulated, downregulated, and non-DEGs, respectively. (**C**) Heatmap showing the expression levels of DEGs in PKCθ-knockout PEDV-infected HEK293T cells (MOI = 1) and PEDV-infected HEK293T cells (MOI = 1) at 48 hpi. Columns represent individual samples, and rows indicate genes with significant differences in expression between the two groups. (**D**) Heatmap showing the expression levels of DEGs in PKCθ-knockout PEDV-infected L929 cells (MOI = 1) and PEDV-infected L929 cells (MOI = 1) at 48 hpi. Columns represent individual samples, and rows indicate genes with significant expression differences between the two groups. (**E**) KEGG classification and functional annotation of PKCθ-knockout PEDV-infected HEK293T cells (MOI = 1) and PEDV-infected HEK293T cells (MOI = 1) at 48 hpi for cellular processes, environmental information processing, genetic information processing, metabolism, and organismal systems. (**F**) KEGG classification and functional annotation of PKCθ-knockout PEDV-infected L929 cells (MOI = 1) and PEDV-infected L929 cells (MOI = 1) at 48 hpi for cellular processes, environmental information processing, genetic information processing, metabolism, and organismal systems. (**G**) KEGG signaling pathway classification in PKCθ-knockout PEDV-infected HEK293T cells (MOI = 1) and PEDV-infected HEK293T cells (MOI = 1) at 48 hpi. The size of the dots indicates the number of DEGs in the signaling pathway, while the color of the dots corresponds to different q-value ranges, and a q-value closer to zero represents a more significant enrichment. (**H**) KEGG signaling pathway classification in PKCθ-knockout PEDV-infected L929 cells (MOI = 1) and PEDV-infected L929 cells (MOI = 1) at 48 hpi. The size of the dots indicates the number of DEGs in the signaling pathway, while the color of the dots corresponds to different *q*-value ranges, and the *q*-value being closer to zero represents a more significant enrichment.

**Figure 7 ijms-25-03096-f007:**
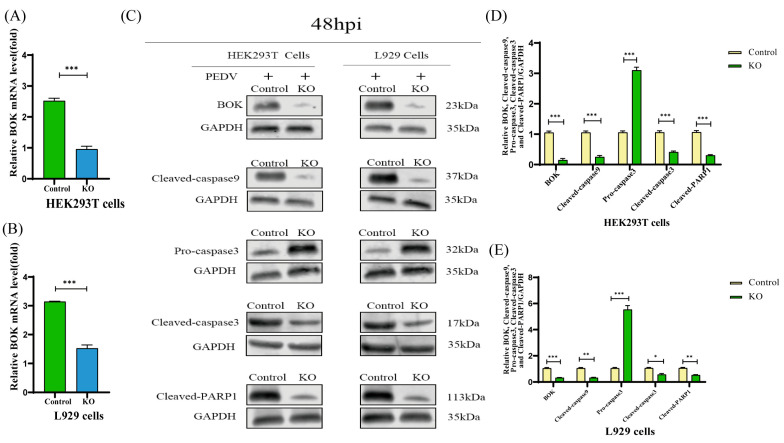
PKCθ knockout downregulates BOK expression and mitochondrial apoptotic signaling. (**A**) qRT-PCR analysis of differential BOK mRNA levels in PKCθ-knockout PEDV-infected HEK293T cells (MOI = 1) and PEDV-infected HEK293T cells (MOI = 1) at 48 hpi. (**B**) qRT-PCR analysis of differential BOK mRNA levels in PKCθ-knockout PEDV-infected L929 cells (MOI = 1) and PEDV-infected L929 cells (MOI = 1) at 48 hpi. (**C**–**E**) Western blot analysis of the protein levels of BOK, cleaved-caspase9, pro-caspase3, cleaved-caspase3, and cleaved-PARP1 in PKCθ-knockout PEDV-infected HEK293T cells (MOI = 1), PEDV-infected HEK293T cells (MOI = 1), PKCθ-knockout PEDV-infected L929 cells (MOI = 1), and PEDV-infected L929 cells (MOI = 1) at 48 hpi; changes in BOK/GAPDH, cleaved-caspase9/GAPDH, pro-caspase3/GAPDH, cleaved-caspase3/GAPDH, and cleaved-PARP1/GAPDH ratios were quantified using ImageJ software version 1.57 and plotted. (**F**) Colocalization analysis of TOM20-stained mitochondria (green), cytochrome c (red), and DAPI-nuclear staining (blue) in PKCθ-knockout PEDV-infected HEK293T cells (MOI = 1) and PEDV-infected HEK293T cells (MOI = 1) to detect cytochrome c leakage. Images were acquired using a Zeiss 880 inverted confocal microscope with a 63× oil-immersion objective. Scale bar, 10 μm. The colocalization of mitochondria (TOM20, green) and cytochrome c (red) was analyzed. (**G**) Colocalization analysis of TOM20-stained mitochondria (green), cytochrome c (red), and DAPI-nuclear staining (blue) in PKCθ-knockout PEDV-infected L929 cells (MOI = 1) and PEDV-infected L929 cells (MOI = 1) to detect cytochrome c leakage. Images were acquired using a Zeiss 880 inverted confocal microscope with a 63× oil-immersion objective. Scale bar, 10 μm. The colocalization of mitochondria (TOM20, green) and Cyt c (red) was analyzed. (**H**) Flow cytometric analysis of the proportion of apoptotic cells in HEK293T cells, PKCθ-knockout PEDV-infected HEK293T cells (MOI = 1), PEDV-infected HEK293T cells (MOI = 1), L929 cells, PKCθ-knockout PEDV-infected L929 cells (MOI = 1), and PEDV-infected L929 cells (MOI = 1) using Annexin V-FITC/PI staining at 48 hpi. The proportion of apoptotic cells was analyzed further. (**I**) Flow cytometry analysis of JC-1 signals in HEK293T cells, PKCθ-knockout PEDV-infected HEK293T cells (MOI = 1), PEDV-infected HEK293T cells (MOI = 1), L929 cells, PEDV-infected L929 cells (MOI = 1), and PKCθ-knockout PEDV-infected L929 cells (MOI = 1) at 48 hpi. The proportion of cells with JC-1 monomers was further analyzed. (**J**) Flow cytometric analysis of the percentage of cells with cleaved caspase-3 in HEK293T cells, PKCθ-knockout PEDV-infected HEK293T cells (MOI = 1), PEDV-infected HEK293T cells (MOI = 1), L929 cells, PEDV-infected L929 cells (MOI = 1), and PKCθ-knockout PEDV-infected L929 cells (MOI = 1) at 48 hpi. The percentage of cells expressing cleaved-caspase3 was further analyzed. “*” means *p* ≤ 0.05; “**” means *p* ≤ 0.01; “***” means *p* ≤ 0.001.

**Figure 8 ijms-25-03096-f008:**
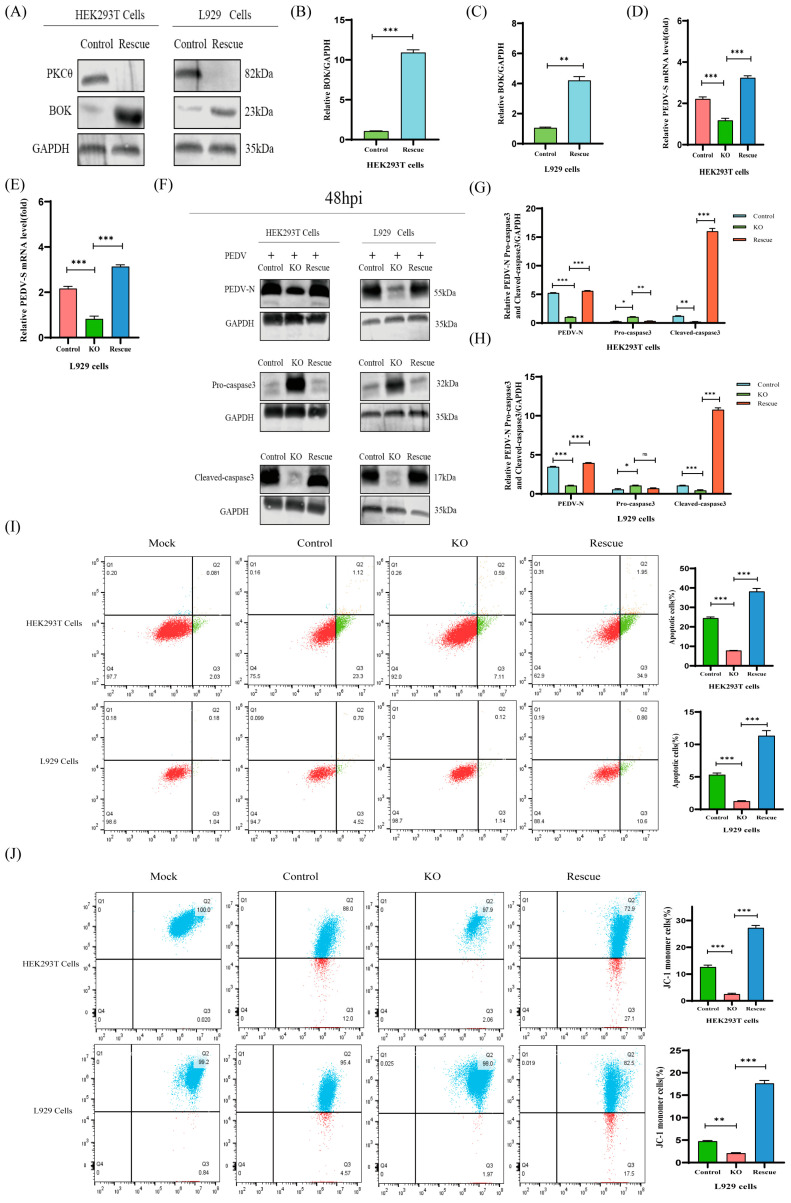
PKCθ–BOK axis-induced mitochondrial apoptosis plays a positive role in PEDV infection. (**A**–**C**) Western blot analysis of the generation of rescued overexpressing BOK in PKCθ-knockout HEK293T and L929 cells; changes in the BOK/GAPDH ratio were quantified using ImageJ software version 1.57 and plotted. (**D**,**E**) qRT-PCR analysis of PEDV-S RNA levels in PEDV-infected HEK293T cells (MOI = 1), PKCθ-knockout PEDV-infected HEK293T cells (MOI = 1), BOK-rescued overexpressing PKCθ-knockout HEK293T cells, PEDV-infected L929 cells (MOI = 1), PKCθ-knockout PEDV-infected L929 cells (MOI = 1), and BOK-rescued overexpressing PKCθ-knockout L929 cells at 48 hpi. (**F**–**H**) Western blot analysis of the protein levels of PEDV-N, pro-caspase3, and cleaved caspase3 in HEK293T cells, PEDV-infected HEK293T cells (MOI = 1), PKCθ-knockout PEDV-infected HEK293T cells (MOI = 1), BOK-rescued overexpressing PKCθ-knockout HEK293T cells, L929 cells, PEDV-infected L929 cells (MOI = 1), PKCθ-knockout PEDV-infected L929 cells (MOI = 1), and BOK-rescued overexpressing PKCθ-knockout L929 cells; changes in PEDV-N/GAPDH, pro-caspase3/GAPDH, and cleaved-caspase3/GAPDH ratios were quantified using ImageJ software version 1.57 and plotted. (**I**) Flow cytometric analysis of the proportion of apoptotic cells in HEK293T cells, BOK-rescued overexpressing PKCθ-knockout PEDV-infected HEK293T cells (MOI = 1), PKCθ-knockout PEDV-infected HEK293T cells (MOI = 1), PEDV-infected HEK293T cells (MOI = 1), L929 cells, BOK-rescued overexpressing PKCθ-knockout L929 cells, PKCθ-knockout PEDV-infected L929 cells (MOI = 1), and PEDV-infected L929 cells (MOI = 1) using Annexin V- FITC/PI staining; the proportion of apoptotic cells was further analyzed. (**J**) Flow cytometry analysis of JC-1 signals in HEK293T cells, BOK-rescued overexpressing PKCθ-knockout PEDV-infected HEK293T cells (MOI = 1), PKCθ-knockout PEDV-infected HEK293T cells (MOI = 1), PEDV-infected HEK293T cells (MOI = 1), L929 cells, BOK-rescued overexpressing PKCθ-knockout L929 cells, PKCθ-knockout PEDV-infected L929 cells (MOI = 1), and PEDV-infected L929 cells (MOI = 1); the proportion of cells with JC-1 monomers was further analyzed. “*” means *p* ≤ 0.05; “**” means *p* ≤ 0.01; “***” means *p* ≤ 0.001; “ns” means not significant.

**Figure 9 ijms-25-03096-f009:**
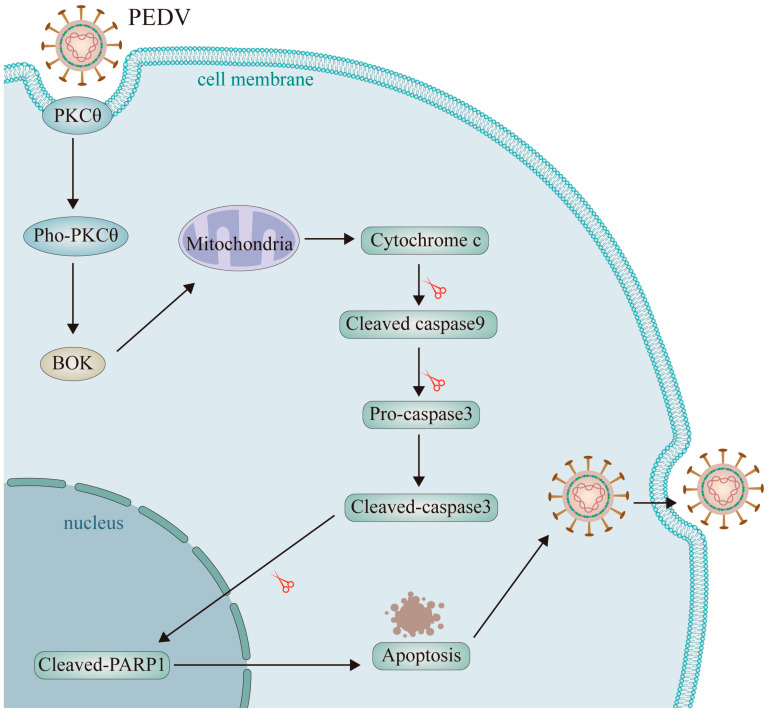
Hypothesis diagram showing that PEDV endocytosis and replication relies on PKCθ-BOK-caspase3 mitochondrial apoptosis.

**Table 1 ijms-25-03096-t001:** Summary of differentially expressed genes (downregulated) in HEK293T and L929 cells determined using RNA-Seq analysis.

Human Gene Stable ID	Mouse Gene Stable ID	Gene Names	Score Ordering
ENSG00000176720	ENSMUSG00000026278	BOK	1
ENSG00000100300	ENSMUSG00000041736	TSPO	2
ENSG00000127663	ENSMUSG00000024201	KDM4B	3
ENSG00000124145	ENSMUSG00000017009	SDC4	4
ENSG00000141959	ENSMUSG00000020277	PFKL	5
ENSG00000100292	ENSMUSG00000005413	HMOX1	6
ENSG00000171388	ENSMUSG00000037010	APLN	7
ENSG00000171227	ENSMUSG00000050777	TMEM37	8
ENSG00000007255	ENSMUSG00000002043	TRAPPC6A	9
ENSG00000173153	ENSMUSG00000024955	ESRRA	10
ENSG00000176171	ENSMUSG00000078566	BNIP3	11
ENSG00000160325	ENSMUSG00000015488	CACFD1	12
ENSG00000072682	ENSMUSG00000018906	P4HA2	13
ENSG00000129355	ENSMUSG00000096472	CDKN2D	14
ENSG00000060971	ENSMUSG00000036138	ACAA1	15
ENSG00000109107	ENSMUSG00000017390	ALDOC	16
ENSG00000111674	ENSMUSG00000004267	ENO2	17
ENSG00000172183	ENSMUSG00000039236	ISG20	18
ENSG00000162522	ENSMUSG00000050390	KIAA1522	19
ENSG00000186352	ENSMUSG00000050914	ANKRD37	20
ENSG00000179403	ENSMUSG00000042116	VWA1	21
ENSG00000136068	ENSMUSG00000025278	FLNB	22
ENSG00000168101	ENSMUSG00000022516	NUDT16L1	23
ENSG00000135622	ENSMUSG00000000627	SEMA4F	24
ENSG00000168273	ENSMUSG00000058351	UQCC5	25
ENSG00000162461	ENSMUSG00000040740	SLC25A34	26
ENSG00000143416	ENSMUSG00000068874	SELENBP1	27

**Table 2 ijms-25-03096-t002:** Summary of differentially expressed genes (upregulated) in HEK293T and L929 cells determined using RNA-Seq analysis.

Human Gene Stable ID	Mouse Gene Stable ID	Gene Names	Score Ordering
ENSG00000198763	ENSMUSG00000064345	MT-ND2	1
ENSG00000228253	ENSMUSG00000064356	MT-ATP8	2
ENSG00000198157	ENSMUSG00000031245	HMGN5	3
ENSG00000205571	ENSMUSG00000021645	SMN2	4
ENSG00000136866	ENSMUSG00000028389	ZFP37	5
ENSG00000172062	ENSMUSG00000021645	SMN1	6

**Table 3 ijms-25-03096-t003:** sgRNAs knockout PKCθ in HEK293T and L929 cells.

sgRNA Names	sgRNA Sequences
sgRNA-1	5′-GACAAGCCAATCCGAAGAAA-3′
sgRNA-2	5′-GGGTGGGTACATGGTAGGCT-3
sgRNA-3	5′-CCTCATCTCAGAAACAACC-3′
sgRNA-4	5′-TGTGGGTTGAGGGAAAAAGG-3′

**Table 4 ijms-25-03096-t004:** Primers used for qRT-PCR.

NCBI Gene ID	Primer Names	Primer Sequences
5588 (PKCθ-human)	5588-RT-F	GCAAAAACGTGGACCTCATCT
	5588-RT-R	CAAAGAAGCCTTCCGTCTCAAA
18761 (PKCθ-mouse)	18761-RT-F	TATCCAACTTTGACTGTGGGACC
	18761-RT-R	CCCTTCCCTTGTTAATGTGGG
935184 (PEDV-S)	935184-RT-F	TGCCAATGTATTTGCCACT
	935184-RT-R	TGACAGTAGGAGGTAAAACAGCC
666 (BOK-human)	666-RT-F	GTCTTCGCTGCGGAGATCAT
	666-RT-R	CATTCCGATATACGCTGGGAC
51800 (BOK-mouse)	51800-RT-F	TGTCTTTGCAGCGGAGATCAT
	51800-RT-R	TCCCGGCCTAGTGCCTTAG
2597 (GAPDH-human)	2597-RT-F	GGAGCGAGATCCCTCCAAAAT
	2597-RT-R	GGCTGTTGTCATACTTCTCATGG
14433 (GAPDH-mouse)	14433-RT-F	AGGTCGGTGTGAACGGATTTG
	14433-RT-R	GGGGTCGTTGATGGCAACA

**Table 5 ijms-25-03096-t005:** Primary antibodies used in our study.

Antibodies’ Names	Catalog. No	Dilution Ratio	Sources
Anti-PKCθ antibody	ab302891	1:1000	Abcam, Cambridge, UK
Anti-phosphorylated PKCθ antibody	T538	1:1000	Cell Signaling, Danvers, MA, USA
Anti-PEDV-N antibody	SD-2-1	1:1000	Medgenes, Brookings, SD, USA
Anti-BOK antibody	ab233072	1:1000	Abcam, Cambridge, UK
Anti Caspase 9 antibody	9509T	1:1000	Cell Signaling, Danvers, MA, USA
Anti-Cleaved Caspase 3 antibody	9664T	1:1000	Cell Signaling, Danvers, MA, USA
Anti-Cleaved PARP1 antibody	ab225715	1:100	Abcam, Cambridge, UK
Anti-GAPDH antibody	60004-1-Ig	1:10,000	Proteintech, Shanghai, China

## Data Availability

Data is contained within the article and Appendix A.

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
