# Peer review of "Unveiling the Role of Protein Kinase C θ in Porcine Epidemic Diarrhea Virus Replication: Insights from Genome-Wide CRISPR/Cas9 Library Screening"

_ijms, 2024, doi:10.3390/ijms25063096_

Round 1
Reviewer 1 Report
Comments and Suggestions for Authors
In this manuscript, Zhou et al. report using Genome-wide CRISPR/Cas9 library screening approach to identify PKCθ is a crucial host factor in PEDV infection. The authors demonstrate that HEK293T and L929 cells are suitable for PEDV infection study, also found PEDV activates the PKCθ-(BCL-2) ovarian killer (BOK) axis to increase viral endocytosis and replication via mitochondrial apoptosis in HEK293T and L929 cells. These results provide import insights about how PEDV infect cells and could be beneficial for developing potential strategies or drug targets for PEDV infection. Overall, the manuscript is easy to follow and is generally well written. I have some Minor Remarks: (1). For some statistical analysis, the authors should clearly describe the significance in the figure, i. e. in figure 1 G and 1H, the significant stars seems to be labeled incorrectly. For figure 3 D, the significance stars seems to be wrong for 24 hpi. The authors should provide all the statistical analysis in supplementary table. (2). For human PKCθ K.O. sgRNA sequence in table 1, the sequence seems be reverse and complementary sequence, the authors should provide target sequence (5' --- 3' NGG). (3). Since PEDV can infect HEK293T and L929 cells via PKCθ, the authors should check the expression of PKCθ in PEDV-infection-limited cell lines like Vero or IPEC-J2 cells, also check whether overexpression of PKCθ in these cell lines could restore the infection of PEDV. (4). If PKCθ is conserved from mouse to human, the author should do phylogenetic analysis, which is very helpful to understand the potential susceptibility to cross-Species infection for PEDV. (5). The Materials and Methods section of the paper requires substantial improvement. Experimental details are either missing or vaguely explained (e.g. exact number of cells in each experiment, exact transfection conditions, origin of cell lines). To enhance the paper's quality, it is crucial to provide a comprehensive and detailed methodological section.
Comments on the Quality of English Language
Overall, the manuscript is easy to follow and quite well written, although it would benefit from editing by a native English speaker.
Author Response
Dear Reviewer 1:
Thank you for your comments concerning our manuscript entitled “Unveiling the Role of PKCθ in PEDV replication: Insights from Genome-wide CRISPR/Cas9 Library Screening” (ID: ijms-2872561). Those comments are all valuable and very helpful for us to improve the quality of our manuscript. I have rigorously revised our manuscript according to your professional comments and marked with blue color. In the next, please allow us to explain our revisions one by one according to your comments.
Comment 1: Minor editing of English language required.
Answer to comment 1: we have carefully checked our manuscript and revised the linguistic problems. In addition, we also used MDPI English editing service to improve the fluency and coherence of our manuscript.
Comment 2: In this manuscript, Zhou et al. report using Genome-wide CRISPR/Cas9 library screening approach to identify PKCθ is a crucial host factor in PEDV infection. The authors demonstrate that HEK293T and L929 cells are suitable for PEDV infection study, also found PEDV activates the PKCθ-BOK axis to increase viral endocytosis and replication via mitochondrial apoptosis in HEK293T and L929 cells. These results provide import insights about how PEDV infect cells and could be beneficial for developing potential strategies or drug targets for PEDV infection. Overall, the manuscript is easy to follow and is generally well written.
Answer to comment 2: I am very appreciating for your conclusion about our manuscript, which encourage me to work hard in exploring the mysteries of virology. In the future, we will endeavor to generate high-quality academic outputs and hope to publish in the journal “international journal of molecular sciences”.
Comment 3: For some statistical analysis, the authors should clearly describe the significance in the figure, i. e. in figure 1 G and 1H, the significant stars seems to be labeled incorrectly. For figure 3 D, the significance stars seems to be wrong for 24 hpi. The authors should provide all the statistical analysis in supplementary table.
Answer to comment 3: I have checked the statistical analysis of figure 1G, figure 1H and figure 3D. I feel very sorry about our incorrect in these statistical analysis. I have re-analyzed the statistics of in figure 1 G and 1H as well as 24hpi in figure 3D again by using ImageJ and Prism 8. Subsequently, I redrew and rearranged these figures regarding to the statistical analysis by using Adobe illustrator. Please check, thank you very much.
Comment 4: For human PKCθ-KO sgRNA sequence in table 1, the sequence seems be reverse and complementary sequence, the authors should provide target sequence (5' --- 3’ NGG).
Answer to comment 4: I feel very sorry about our negligence of using 3’-5’ sgRNA sequence for human PKCθ-KO. We have revised it to 5’-3’ sgRNA sequence and marked with blue color. Please check, thank you very much.
Comment 5 and 6: Since PEDV can infect HEK293T and L929 cells via PKCθ, the authors should check the expression of PKCθ in PEDV-infection-limited cell lines like Vero or IPEC-J2 cells, also check whether overexpression of PKCθ in these cell lines could restore the infection of PEDV. If PKCθ is conserved from mouse to human, the author should do phylogenetic analysis, which is very helpful to understand the potential susceptibility to cross-Species infection for PEDV.
Answer to comment 5 and 6: Please allow me to give you a reasonable explanation why we not validate the expression of PKCθ in PEDV-infected Vero and IPEC-J2 cells as well as phylogenetic analysis. Firstly, we constructed Genome-wide CRISPR/Cas9 library screening in HEK293T cells and found PKCθ can be recognized as host factors required for PEDV infection in HEK293T cells and L929 cells in purpose of preliminarily examining the cross-species infectivity of PEDV. Secondly, previous research (PMID: 28470417) has illustrated HEK293 can be considered as modeling cells to do research on PEDV. Thirdly, another research (PMID: 37499712) identified a different set of genes required for PEDV infection in vero cells, raising questions about potential discrepancies due to different host cells. To conclude, we add these questions in the discussion part, which can give instructions and suggestions to other researchers who are majored in doing research on the PEDV infection mechanisms. I marked these changes in discussion part with blue color, thank you very much.
Comment 7: The Materials and Methods section of the paper requires substantial improvement. Experimental details are either missing or vaguely explained (e.g. exact number of cells in each experiment, exact transfection conditions, origin of cell lines). To enhance the paper's quality, it is crucial to provide a comprehensive and detailed methodological section.
Answer to comment 7: Thanks you for pointing this out. I have re-written the materials and methods, giving an exactly and detailed methodology of this manuscript. I marked the revised part in materials and methods with blue color, please check, thank you very much.
In conclusion, as the first author who do the majority work of this manuscript, I am very appreciating for your distinguishing comments of this manuscript, which help us a lot to improve this manuscript. This manuscript determines whether I can be graduated in this year June or not, so your kindly help and professional comments of this manuscript can aid me to graduate smoothly.
Best wishes!
Mr. Jinglin Zhou
E.mail: qsx20211380@stdent.fjnu.edu.cn
2024.2.22
Reviewer 2 Report
Comments and Suggestions for Authors
In the manuscript “Unveiling the Role of PKCθ in PEDV Replication: Insights from Genome-wide CRISPR/Cas9 Library Screening” Zhou et al., tried to use Genome-wide CRISPR/Cas9 Library Screening methodology combine with RNA-seque to explore the host factors for the PEDV infection. The results are convinced and scientific sound. However, some critical issues should be clarified before published
Major comments:
1. The fold of infection should be indicated in the results sections.
2. PKCθ is not the highest score in the Genome-scale CRISPR screens (Fig. 2B). As the authors indicated “relatively high” but lower than OR52M1, GPI, MGEA5 …, so, why not test OR52M1 and devoted to PKCθ should be give an convincing explanations. And, what are the functions of the OR52M1, GPI, MGEA5 genes encoded proteins?
3. In Figure 3 G and I, there were smear bands in the phosphorylated PKCθ, and two bands can be identified, roughly. The molecular weights indicated is 79 kDa, which band is the right one? Furthermore, the molecular weight of PKCθ is 82 KDa higher than the phosphorylated PKCθ, do the results consist with previous results?
Minor comments:
1. high-throughput DNA sequencing (NGS)? NGS is next generation sequencing?
2. To validate the host factor necessary for PEDV infection, we used extra L929 cells in our study. The English is not clear, what is the extra L929 cells? More than common used?
3. The labels in the Y-axis are not clear? The symbol is not clear indicated.
4. Line248: ….FDR ≤ 0.05, What is FDR?
Comments on the Quality of English LanguageThe english should be improved.
Author Response
Dear Reviewer 2:
On behalf of all the contributing authors, I would like to express our sincere appreciations of your constructive comments concerning our article entitled “Unveiling the Role of PKCθ in PEDV replication: Insights from Genome-wide CRISPR/Cas9 Library Screening” (ID: ijms-2872561). These comments are all valuable and helpful for improving our article. According to your comments, we have made extensive modifications in our manuscript to make our results convincing. In this revised version, changes to our manuscript were all highlighted within the document by using green-colored text. Point-by-point responses to the comments or suggestions of reviewer 2 are listed below this letter.
Comment 1: Extensive editing of English language required
Response to comment 1: we have carefully checked our manuscript and revised the linguistic problems. In addition, we also used MDPI English editing service to improve the fluency and coherence of our manuscript.
Comment 2: In the manuscript “Unveiling the Role of PKCθ in PEDV Replication: Insights from Genome-wide CRISPR/Cas9 Library Screening” Zhou et al., tried to use Genome-wide CRISPR/Cas9 Library Screening methodology combine with RNA-seque to explore the host factors for the PEDV infection. The results are convinced and scientific sound.
Response to comment 2: I am very appreciating for your remarks of our article, which give me faith to do scientific research in the future. It is my pleasure to receive such a positive feedbacks from reviewer, I will endeavor myself to solve the scientific problems in my area, and hope to publish high-quality articles or reviews in the journal “international journal of molecular sciences” in the future.
Comment 3: The fold of infection should be indicated in the results sections
Response to comment 3: we have carefully checked the results sections where we have not described the fold of PEDV infection, and then indicated it. We marked these changes with green colors, please check. Thank you very much.
Comment 4: PKCθ is not the highest score in the Genome-scale CRISPR screens (Fig. 2B). As the authors indicated “relatively high” but lower than OR52M1, GPI, MGEA5 …, so, why not test OR52M1 and devoted to PKCθ should be give an convincing explanations. And, what are the functions of the OR52M1, GPI, and MGEA5 genes encoded proteins?
Response to comment 4: As the figure shows, the score of PKCθ is not the highest, and the score of OR52M1, GPI, and MGEA5 and so on is higher than PKCθ. Please allow me to give a reasonable explanation.
Firstly, I will introduce these genes and their encoding proteins’ information to you (all information from gene cards).
1.OR52M1: OR52M1 (Olfactory Receptor Family 52 Subfamily M
Member 1) is a Protein Coding gene. Among its related pathways are Olfactory Signaling Pathway. Gene Ontology (GO) annotations related to this gene include G protein-coupled receptor activity and olfactory receptor activity. An important paralog of this gene is OR52R1.
2. GPI: GPI (Glucose-6-Phosphate isomerase) is a Protein Coding gene.
Diseases associated with GPI include Hemolytic Anemia, Nonspherocytic, due to Glucose Phosphate Isomerase Deficiency and Hemolytic Anemia. Among its related pathways are glycolysis (BioCyc) and Gene expression (Transcription). Gene Ontology (GO) annotations related to this gene include cytokine activity and monosaccharide binding.
3.MGEA5: MGEA5 is the aliases of OGA gene. OGA (O-GlcNAcase) is
a Protein Coding gene. Diseases associated with OGA include Danubian Endemic Familial Nephropathy and Gm2 Gangliosidosis. Among its related pathways are protein O-[N-acetyl]-glucosylation.
Secondly, I select Prkcq-PKCθ to validate whether it is the host factor required for PEDV infection because the other genes, such as OR52M1, GPI and MGEA5 and their underlying mechanism under PEDV infection are examined by my colleagues. These data should be keep confidentiality to some extent before publication because these data determine whether they can be graduated or not. I hope you can understand these, I am very appreciating for this.
Thirdly, among these host factors enriched from GECKO, the PKCθ-KO HEK293T and L929 cells can restrict PEDV infection far more than other host factors. This is the concrete evidence why I choose PKCθ for further validation.
Comment 5: In Figure 3 G and I, there were smear bands in the phosphorylated PKCθ, and two bands can be identified, roughly. The molecular weights indicated is 79 kDa, which band is the right one? Furthermore, the molecular weight of PKCθ is 82 KDa higher than the phosphorylated PKCθ, do the results consist with previous results?
Response to comment 5: we used the primary antibody “Catalog. No: T538” from CST Company for examine the phosphorylation of PKCθ. In the detection of phosphorylated protein, the lower specificity of primary antibody can cause the smear bands. As the figures shows, the band in the top is the phosphorylated PKCθ, and the below one is smear band. In addition, the predicted molecular weight of phosphorylated PKCθ is approximately 79kDa and the predicted molecular weight of PKCθ is 82kDa. These information was acquired from CST website and abcam website. In our WB experiment, we also validate this. Please check, thank you very much.
Comment 6: high-throughput DNA sequencing (NGS)? NGS is next generation sequencing?
Response to comment 6: Yes, NGS is the next generation sequencing. We have write the full name of NGS in the manuscript and marked with green color, please check, thank you.
Comment 7: To validate the host factor necessary for PEDV infection, we used extra L929 cells in our study. The English is not clear, what is the extra L929 cells? More than common used?
Response to comment 7: In this sentence, we want to express that we not only validate the mechanism of host factor PKCθ under PEDV infection in HEK293T cells,but also examine in L929 cells too. The purpose of this sentence was to tell readers that we use two different cell types to increase the credibility of results. Now we changed this sentence in a logistic form and marked changes with green colors, pleases check, thank you.
Comment 8: The labels in the Y-axis are not clear? The symbol is not clear indicated.
Response to comment 8: we have changed the description of Y-axis of Figure.2B, and redrew the figure 2. Please check, thank you very much.
Comment 9: .FDR ≤ 0.05, what is FDR?
Response to comment 9: FDR is the abbreviation of false discovery rate, when FDR ≤ 0.05, the data is statistically significant. In addition, we have added the full name of FDR in line 248 and marked with green color, please check, thank you very much.
In conclusion, as the first author who do the majority work of this manuscript, I am very appreciating for your distinguishing comments of this manuscript, which help us a lot to improve this manuscript. This manuscript determines whether I can be graduate or not, and whether I can pursue my PH.D study. So your kindly help and professional comments of this manuscript can aid me a lot.
Best wishes!
Mr. Jinglin Zhou
E.mail: qsx20211380@stdent.fjnu.edu.cn
2024.2.22